# The Role of Wnt Signalling in Chronic Kidney Disease (CKD)

**DOI:** 10.3390/genes11050496

**Published:** 2020-04-30

**Authors:** Soniya A. Malik, Kavindiya Modarage, Paraskevi Goggolidou

**Affiliations:** Department of Biomedical Science and Physiology, Faculty of Science and Engineering, University of Wolverhampton, Wulfruna Street, Wolverhampton WV1 1LY, UK; s.a.malik2@wlv.ac.uk (S.A.M.); k.modarage@wlv.ac.uk (K.M.)

**Keywords:** Wnt signalling, polycystic kidney disease, nephronophthisis, beta-catenin, planar cell polarity signalling

## Abstract

Chronic kidney disease (CKD) encompasses a group of diverse diseases that are associated with accumulating kidney damage and a decline in glomerular filtration rate (GFR). These conditions can be of an acquired or genetic nature and, in many cases, interactions between genetics and the environment also play a role in disease manifestation and severity. In this review, we focus on genetically inherited chronic kidney diseases and dissect the links between canonical and non-canonical Wnt signalling, and this umbrella of conditions that result in kidney damage. Most of the current evidence on the role of Wnt signalling in CKD is gathered from studies in polycystic kidney disease (PKD) and nephronophthisis (NPHP) and reveals the involvement of β-catenin. Nevertheless, recent findings have also linked planar cell polarity (PCP) signalling to CKD, with further studies being required to fully understand the links and molecular mechanisms.

## 1. Chronic Kidney Disease (CKD)

Chronic kidney disease (CKD) is a term that denotes the presence of kidney damage, presenting with abnormalities or a decline in kidney function. In order to distinguish between chronic and acute kidney disease as well as assess the rate of decline in kidney function, kidney damage is measured over the course of three months via the glomerular filtration rate (GFR) and it can be split across five stages, depending on kidney function efficiency [1]. The first stage refers to a normal or high GFR of > 90 mL/min [1], while the second stage refers to the initial display of an abnormal GFR, which is slightly decreased (GFR 60–89 mL/min) [1]; this can be defined as the mild stage of CKD. The third stage is a more moderate display of CKD, with the GFR being around half of the normal GFR (30–59 mL/min) [1]. Stage four is a severe presentation of CKD and the GFR in this stage is around 15–29 mL/min. In comparison, stage five describes end-stage renal disease (ESRD), where the GFR is <15 mL/min [1]. A number of kidney conditions, both inherited and acquired, have been associated with reduced GFR and are a result of various genetic and environmental factors. The particular focus of this review is to discuss the role of Wnt signalling in representative genetically inherited kidney conditions and shed light on its significance, hoping to utilise this knowledge to inform personalised medicine approaches.

## 2. The Clinical Symptoms of Genetically Inherited Chronic Kidney Diseases

Genetically inherited kidney diseases are an umbrella of various conditions that result from mutations in genes that regulate kidney function (Table 1). These can be of dominant or recessive inheritance and can manifest in individuals from a relatively young age. The most common of these conditions is autosomal dominant polycystic kidney disease (ADPKD). ADPKD is an inherited disorder that affects around 12.5 million people in the world with no ethnic, racial or gender bias [2]. The disease itself is one of the most common causes of end-stage renal disease (ESRD), with statistics showing that ADPKD accounts for around 10% of all ESRD cases [2]. Symptoms of the disease include the presentation of renal and hepatic cysts, together with hypertension, gross haematuria, nephrolithiasis, urinary tract infection, shortness of breath and a general discomfort in the lower back [3]. Whilst children with the disease are usually asymptomatic, symptoms of the disease tend to manifest in adults between the ages of 30 and 50 [3]. ADPKD has been associated with mutations in two genes—*PKD1* and *PKD2*—either of which encode for Polycystin-1 (PC-1) and Polycystin-2 (PC-2), respectively. Evidence has suggested that *PKD1* mutations account for the majority of ADPKD cases, however, patients with mutations in *PKD2* are believed to have a better prognosis [4,5]. Unfortunately, no pharmacological cure currently exists for ADPKD although a recent drug, Tolvaptan, has been shown to slow down the progression of cysts [2].

Autosomal Recessive Polycystic Kidney Disease (ARPKD) is a rare, genetic disorder with an incidence rate of 1/20,000 [6]. It is a disease most common in neonates and infants, equally affecting both boys and girls [6]. The disease manifests with a variety of abnormalities including bilateral enlargement of both kidneys, presenting with fluid-filled cysts throughout the collecting ducts of the kidney, oligohydramnios, hepatic fibrosis and respiratory insufficiency [6,7]. Consequently, patients suffer from renal failure, hypertension, portal hypertension and pulmonary hypoplasia, with death arising in around 30% of affected neonates [6]. ARPKD has been associated with mutations in *Polycystic Kidney and Hepatic Disease 1 (PKHD1)*, a gene encoding for a large, single-transmembrane protein, Fibrocystin. Fibrocystin has been found to be localised to the primary cilium and basal body in the kidneys, with expression also being observed in other organs, including the liver and lungs [8,9]. Recent evidence has also identified *DZIP1L1* as a second gene associated with ARPKD, localised to the centrioles and at the distal end of the basal body of the primary cilium [10].

Nephronophthisis (NPHP) is another autosomal recessive cystic kidney disease that is a leading cause of ESRD in children and young adults [11]. The disease itself presents with symptoms such as polyuria, polydipsia, anaemia, growth retardation and hypertension with characteristics including reduced kidney size, the development of cysts in the corticomedullary area and loss of corticomedullary differentiation [11,12]. NPHP can be categorised into three different forms, including juvenile NPHP, which is the most common form of the disease, where patients tend to reach ESRD by the age of around 13; infantile NPHP, where patients reach ESRD before the age of 4; and adolescent NPHP where the onset of ESRD is around 19 years of age [13,14,15]. Besides this, the diagnosis of NPHP is dependent on the results observed in renal biopsies (including the presence of tubular atrophy, interstitial fibrosis, thickening and attenuating of tubular basement membranes) and genetic testing [12]. To date, up to 20 *NPHP* genes have been implicated in the disease—the most common being *NPHP1,* encoding Nephrocystin-1 and *NPHP2*, encoding Inversin (Inv) [16]. Mutations in some of these *NPHP* genes, including in *NPHP6/CEP290,* have been associated with other syndromes including Joubert syndrome (JS) and Meckel–Gruber syndrome (MGS) with evidence displaying that around 20%–30% of JS patients also develop NPHP [16,17,18,19]. JS is characterised by hypotonia, hyperpnea, abnormal eye movements, delays in developmental abilities and ptosis. When presented with additional symptoms including kidney disease, liver disease and skeletal abnormalities, the disease is referred to as ‘Joubert syndrome and related disorders (JSRD) [20]. In comparison, MGS presents with symptoms including polycystic kidneys, polydactyly and occipital encephalocele with 100% mortality rate [21]. Both JS and MGS are inherited in an autosomal recessive pattern and have been categorised alongside ADPKD, ARPKD and NPHP as ‘ciliopathies’, a term which denotes defects in primary cilia [20,21]. Primary cilia have been implicated in kidney development and disease and are linked to proteins that are associated with cystic renal diseases, including the diseases mentioned above [22]. Signalling via the primary cilium is also thought to be a crucial process and evidence has found that defects in cilia can impact cilia-associated signalling pathways, including Wnt signalling [23].

IgA nephropathy (IgAN) is one of the most common forms of glomerulonephritis and another leading cause of CKD and ESRD, with an incidence rate of 2.5/100,000 [24]. Clinical manifestations of the disease are variable with common presentations including microscopic/macroscopic haematuria, together with the presentation of proteinuria [25]. Another common characteristic is synpharyngitic macroscopic haematuria, where episodic haematuria follows an upper respiratory tract infection [25]. The diagnosis of IgAN is dependent on immunofluorescent analysis on kidney biopsy samples, where granular deposition of IgA in mesangium is usually observed [25]. Despite the continuous research trying to establish the cause and genetic basis of IgAN, there is no definitive causative gene(s) that has been established to date, rather indications of genetic factors involved in the disease [26]. Varying prevalence of IgAN has been observed in different ethnic groups, with a higher prevalence of IgAN found in Asian populations compared to Europe and North America. In addition, in Europe, there is higher prevalence of IgAN in men than women and an increased risk of IgAN in relatives of patients in Europe—this is not observed in Asia [26,27]. It is key to note that there may be a limitation in this finding, due to differences in the criteria for the use of renal biopsies across different geographical locations. Recently, there has been an increase in renal biopsy use in Europe, which may account for the increase in IgAN prevalence observed [26]. Despite this, genome-wide association studies in European and South-East Asian populations have highlighted risk alleles in the HLA region at chromosome 6p21 and chromosome 1q32 [28].

Focal and segmental glomerulosclerosis (FSGS), a common cause of nephrotic syndrome, refers to the presentation of scarring on certain parts of the glomeruli, whilst other parts remain unaffected [29]. In the US, the incidence rate has been reported at around 7/1,000,000, with the number of ESRD cases being accounted for by FSGS relatively increasing, whilst McGrogan et al. reported the incidence rate for FSGS as around 0.8/100,000 per annum, worldwide [24,30]. Clinical symptoms of the disease include oedema, proteinuria, microscopic haematuria, hypalbuminaemia and hypertension, however, a definitive diagnosis is dependent on analysis of a kidney biopsy sample [29]. FSGS can be sub-categorised based on a classification system that considers the underlying cause of the disease. This includes idiopathic (primary) FSGS and secondary FSGS, which account for genetic/virus-associated forms [31]. Despite this, Rosenberg and Kopp have categorised FSGS in a similar manner into six different forms including primary FSGS, adaptive FSGS, *APOL1* FSGS, genetic FSGS, infection/inflammation-associated FSGS and medication-associated FSGS, whilst the disease can also be classified based on the morphological appearance of FSGS lesions (perihilar, cellular, tip, collapsing and not otherwise specified) [29,32,33,34]. Several factors are believed to be involved in the pathogenesis of the disease, which can vary, but the result of these factors is damage to podocytes/podocyte lesions, consequently resulting in the phenotypic characteristics presented in FSGS [35]. Current research is still working on identifying an underlying genetic mechanism of the disease, with recent findings in African descent patients associating FSGS with variants in *APOL1* [36].

## 3. The Role of Upstream Wnt Components in Genetic CKD

In recent years, there has been an accumulation in the number of studies implicating Wnt ligands and components of the Wnt signalling pathway with CKD (Table 2). Immunohistochemical evidence of kidney biopsy samples has highlighted the presence of several Wnt proteins in common CKDs, including in IgAN and FSGS [37]. The majority of Wnt ligands expressed in the fibrotic kidney, including Wnt1, Wnt2, Wnt4, Wnt5a, Wnt9b and Wnt10b, were localised in the tubular epithelium [37]. Renal tubule specific inhibition of Wnt secretion resulted in the inhibition of fibroblast gene expression activation, as well as the improvement in kidney fibrosis, and thus it was concluded that there may be a role for tubule-derived Wnt proteins in fibroblast activation and kidney fibrosis [37]. It is key to note that some research has suggested that certain Wnt ligands are predominantly activated in either the canonical or non-canonical Wnt signalling pathway, however, there has been evidence associating Wnt ligands, such as Wnt3a and Wnt5a, to both signalling pathways [38,39,40,41].

One of the key Wnt ligands that has an established role in the pathogenesis of CKD and is believed to predominantly act through the canonical Wnt signalling pathway is Wnt1 [38]. Published data have suggested that the overexpression of Wnt1 and active β-catenin is specifically associated with FSGS and diabetic nephropathy (DN) in glomerular podocytes [42]. This comes from evidence whereby the expression of Wnt1 and active β-catenin was observed in DN and FSGS kidneys, whilst that expression was not detected in normal human kidney glomeruli [42]. Further implicating Wnt1 and the canonical Wnt signalling pathway to CKD is the role they play in podocyte dysfunction and the exacerbation of albuminuria [42]. The upregulation of Wnt1 *in vivo*, which activated glomerular β-catenin, as well as the pharmacologic activation of β-catenin, seemed to induce albuminuria, whilst also presenting with podocyte lesions [42]. To strengthen this finding, upon treatment with an antagonist of Wnt signalling, Dickkopf (DKK1), the presentation of podocyte injury and albuminuria improved [42,57]. DKK proteins antagonise Wnt signalling by binding and internalising low-density lipoprotein-receptor-related protein 5 (LRP5) or 6 (LRP6) [57].

Besides Wnt1, other Wnt ligands have been shown to play a role in kidney development and have a known involvement in genetic kidney diseases. One such example is Wnt4, which is required for the conversion of metanephric mesenchyme (MM) to epithelia that form the nephron. Wnt4-induced tubule formation in MM cells and studies in genetically modified mice showed Wnt4 expression in pretubular aggregates [58]. Wnt4-induced tubulogenesis was mediated by the non-canonical Wnt/calcium pathway and an influx of Ca^2+^ occurred when MM cells were induced by Wnt4 [58]. Wnt4 expression was observed as early as stage 25 and was halted at stage 35 in *Xenopus* embryos upon the formation of epithelial structures [59]. In *Pkd1^−/−^* mice, Wnt4 expression was rapidly downregulated by embryonic day (E)14.5 and at E15.5 renal and pancreatic cysts had developed [43]. There is also evidence that the transcription factor GLIS2 regulates Wnt4 expression, with homozygous mutant *GLIS2^-^/^-^* mice and *GLIS2*-knockdown IMDC3 cells showing upregulated Wnt4 expression and *GLIS2* mutations linked to Nephronophthisis 7 in humans and mice [60]. A recent study performed by Kiewisz et al. investigated Wnt4 expression in several glomerular diseases and most importantly at various stages of CKD [44]. WNT4 mRNA and protein expression was absent in normal adult kidneys, until renal injury or onset of kidney diseases caused the activation of WNT4 expression. A consistent increase in *WNT4* expression was observed in all CKD stages. In addition, a small significant decrease was detected between stage 2 and 3. As well as this, the greatest WNT4 expression was seen in membranous nephropathy compared to Lupus nephritis and IgAN/FSGS [44].

Wnt5a is another Wnt ligand with demonstrated functions in the kidney. Bilateral duplex kidneys were observed in *Wnt5a fl/+;T-Cre* and *Wnt5a fl/Δ;T-Cre* mice along with shortened tail, limbs, digits, face, and genitals [61]. Increased Wnt5a expression in zebrafish led to kidney cyst formation in all nephron structures, signifying Wnt5a’s importance for correct kidney morphogenesis [62]. A recent study demonstrated that the upregulation of *WNT5A* expression was noted in ARPKD compared to age-matched normal paediatric kidneys, proposing its novel involvement in ARPKD through interactions of Fibrocystin with ATMIN, an effector PCP transcription factor [45]. A recent clinical study assessed vascular calcification in 41 patients with ESRD and revealed an upregulation in WNT5A and β-CATENIN expression in radial arteries obtained during arteriovenous fistula surgery, concluding that expression of *WNT5A* was an independent risk factor that could result in vascular calcification [46]. To further this, humans and mice exhibiting loss of function (LOF) of *WNT5A* presented multiple-organ defects and patients harbouring the most prominent *WNT5A* LOF variants all exhibited kidney malformation [63].

Wnt9b is a crucial Wnt ligand that plays a role in both canonical Wnt signalling and non-canonical Wnt/PCP signalling. This comes from evidence that displays that mis-expression of Wnt9b in Six2-positive cells disrupts kidney function by activating canonical Wnt signalling in *Wnt9b* transgenic mice [48]. As well as this, researchers highlighted the upregulation of canonical Wnt signalling, together with the presence of renal cysts throughout the nephron, further supporting the idea that disruption of canonical Wnt signalling can result in cystic disease [48]. Despite this, other evidence has also highlighted that regulation of Wnt9b is crucial for PCP in the epithelium of the kidney and that depleted *Wnt9b* resulted in an increased diameter in tubules [49]. Similarly, attenuation of Wnt9b resulted in renal cystogenesis [49].

Wnt11 binds to Frizzled (Fz) 4 to induce canonical and non-canonical signalling in the same cell line (human embryonic kidney 293 (HEK293)), whereas the binding of Wnt11 and Fz8 induces the non-canonical signaling pathway only [64]. In *Wnt11^tm1a/tm1a^* mice, the tubular morphology was disrupted, and glomerular cysts were present. The disruption in tubular morphology caused the population of normal nephrons in *Wnt11^-/-^* mice to be reduced by half at 6 weeks post birth [50]. Additionally, a noticeable increase in the size of the glomeruli and renal capsules was observed, along with a 28% decrease in kidney size [50]. Examination of C57BL/6 *Wnt11^-/-^* mice revealed that the kidneys were hypoplastic and small cysts had formed by E16.5. Decreased cell population and reduced expression of stromal progenitor markers, e.g., *Hox10* and *Foxd1* were also observed in *Wnt11^-/-^* mice at E16.5 [65]. It was postulated that the absence of Wnt11 in *Wnt11^-/-^* mice was believed to trigger the downregulation of the following genes: *Six2, Wnt9b, Gdnf, and Foxd1* [50]. The conservation of renal architecture in a smaller kidney volume was similarly observed in *Wnt11^-/-^* embryos and Fz4^-/-^ /Fz8^-/-^ embryos [64].

## 4. Wnt Signalling Receptor Involvement in CKD

Wnt signalling activation results after a Wnt ligand binds to a Fz receptor, sometimes also in the presence of other co-receptors. To date, there are a total of 10 identified mammalian Fz genes, with expression of all these being observed in the human kidney [66,67,68,69,70,71,72,73]. A 30% decrease in kidney size was observed in homozygous mutant *Fz4* and *Fz8* mice, whereby renal architecture was preserved with no evidence of cyst formation, but a reduction in ureteric bud branching [64]. A significant upregulation of Fz3 and downstream effector CDC42 was observed in *Pkd1* knockout (KO) cystic mouse kidneys [74]. Fz3 expression increased before or simultaneously with the rate of cyst formation and it was postulated that the increase in Fz3 contributed to the growth and enlargement of the cysts [74].

LRP5/LRP6 are crucial co-receptors of the canonical Wnt signalling pathway, since its activation is dependent on the binding of a canonical Wnt ligand to a Fz receptor in the presence of these co-receptors [75]. Mutations in *LRP5* have been linked to hepatic cystogenesis and may be associated with ADPKD, further emphasising the role of the key canonical Wnt signalling co-receptors LRP5/LRP6 in cystogenesis and PKD [51,76]. Cnossen et al. observed four *LRP5* variants in sporadic ADPKD patients, whilst reduced activation of canonical Wnt signalling and altered Wnt target gene expression was detected upon transfection of HEK293 cells with mutant LRP5 [51,76]. This included increased gene expression of target genes including adenomatous polyposis coli (*APC*), glycogen synthase kinase (*GSK3β)*, axis inhibitor-1/2 (*AXIN-1/AXIN-2*) and *c-Myc,* whilst, in the presence of Wnt3a, decreased gene expression was observed for *GSK3β*, *AXIN-1*, *AXIN-2* and *c-Myc* [76]. Furthermore, mutant mouse embryos whereby an insertion mutation links the first 321 amino acids of the LRP6 protein with the β*geo* reporter gene have been demonstrated to be embryonic lethal and result in polycystic kidneys [77]. In addition to this, LRP6 KO in another mouse study displayed similar results [52]. At E11.5, fewer mesonephric tubules were detected in LRP6 KO mice in comparison to littermate controls, and at E18.5, all LRP6 KO mouse kidneys were hypoplastic and exhibited several cysts [52]. Recent evidence has also highlighted an interaction in PKD between LRP6 and *AHI1,* a gene that encodes for the protein Jouberin (Jbn) and that has been associated with NPHP [52,78]. Jbn is believed to interact with and facilitate β-catenin nuclear accumulation, whilst loss of the protein resulted in a decrease in β-catenin function [52,78], with Jbn and LRP6 potentially sharing the same signalling pathway, since renal atrophy and renal cysts were seen in heterozygous *Ahi1^+/-^*; *LRP6^+/-^* and homozygote *Ahi1^-/-^* mice [79].

Aside from canonical Wnt receptors, core PCP receptors that are evolutionarily conserved from *Drosophila* to mammals have been demonstrated to have a crucial involvement in the kidney. Conserved PCP proteins have further been reported to influence the function of cilia, hence why mutations in PCP genes have been shown to cause ciliopathies ranging from PKD to rarer genetic disorders like MGS, Bardet–Bieldl syndrome and NPHP [80]. It has been shown that Vangl2 is normally expressed in epithelial podocytes that regulate normal differentiation, which later induces the formation of the glomeruli, nephron tubules and collecting duct tubules [81]. Point mutation of *Vangl2* in homozygous *loop-tail (Lp)* mice resulted in malformed kidneys alongside dysmorphic glomeruli [81]. Immature glomeruli and hypoplastic medulla were observed in *Vangl2^Lp/Lp^* at E17.5; interestingly, heterozygous *Vangl2^Lp/+^* mouse organs appeared normal and retained morphological structures. Immunostaining of *Vangl2^Lp/Lp^* fetal kidneys revealed the presence of primary cilia, implying that the absence of Vangl2 does not affect ciliogenesis [82]. Recent work has revealed a genetic interaction between Vangl2 and Atmin [45]. Immunostaining of E13.5 *Atmin^Gpg6/Gpg6^* embryonic kidneys revealed a downregulation of Vangl2 expression, although *Vangl2* mRNA levels remained unchanged. In addition, the degree of craniorachischisis in both *Atmin^Gpg6/+^* and *Vangl2*^Lp/+^ mutant mice was noted to be similar, with double heterozygous mice revealing a genetic interaction [45]. An upregulation of VANGL2 expression was observed in human ARPKD samples compared to normal age-matched kidneys [47].

The published literature shows that Celsr1 is strongly involved with epithelial planar cell polarity, whereas Celsr2 and Celsr3 are often associated with ciliogenesis and neural development, respectively [83]. Interestingly, CELSR1-3 show variable expression in the kidneys, lungs, reproductive systems, gastrointestinal and the skin [83]. *Celsr1^crsh/crsh^* mice displayed severely dysmorphic kidneys with dilated tubules, immature glomeruli and immature tubular architecture [84]. To further support the significance of CELSR in relation to the manifestation of diseases, an extensive clinical evaluation of 100 patients from 86 families diagnosed with JS discovered that *CELSR2* was a novel pathogenic variant that was not previously associated with the manifestation of JS, although it only accounted for a small proportion (1%) of patients [56].

Finally, decreased SCRIBBLE expression was observed in ADPKD cell lines and *Pkd2* morphant zebrafish, with loss of Scribble causing pronephric ducts to be dilated [85]. Scribble colocalised with PC2 and this interaction reduced cyst growth in *Pkd2* morphants, revealing a novel role for Scribble in cyst formation in ADPKD via the YAP pathway [85]. Interestingly, an upregulation of *SCRIBBLE* was observed in human ARPKD kidney biopsies; likewise, in *Pkhd1* knockdown mIMDC3 cells, a 60% increase in *Scribble* mRNA expression was detected, suggesting links between Scribble and Fibrocystin in ARPKD [47]. Furthermore, knockdown of Scribble caused the depletion of Fat1, leading to cyst formation during zebrafish pronephros development and suggesting an overlap in signalling cascades [86].

## 5. The Role of β-Catenin in Genetic CKD

Despite playing a crucial role in organ development, canonical Wnt signalling is comparatively inactive in normal adult kidneys [87], but becomes activated after renal injury [88]. Research highlights a protective role for canonical Wnt signalling after acute kidney injury (AKI) however, despite this, evidence has demonstrated that continuous β-catenin activation could become detrimental and lead to CKD [88,89,90]. It has been observed that the role that β-catenin plays in disease is similar to the role it plays in development, whereby although β-catenin is necessary to initiate nephrogenesis, sustained β-catenin action could result in defective epithelialisation [89]. In a study by Xiao et al. where researchers used moderate (20 min) and severe (30 min) ischaemia reperfusion injury (IRI) mice, it was found that the moderate IRI in mice led to acute kidney failure and β-catenin activation, followed by restoration of kidney morphology and function [90]. In comparison, the severe IRI mice displayed continuous β-catenin activation and an eventual appearance of CKD, presented by renal fibrosis [90]. This upregulation of β-catenin has proven to be a pathologic feature in several fibrotic CKDs including obstructive nephropathy, DN, Adriamycin (ADR) nephropathy, remnant kidneys after 5/6 nephrectomy, PKD and chronic allograft nephropathy [42,91,92,93,94,95]. Besides this, an increase in nuclear accumulation of β-catenin has been observed in the peripheral blood leukocytes from IgAN patients, whilst evidence has also displayed the activation of the canonical Wnt signalling pathway in lupus nephritis patients [53,96]. This presence of β-catenin activation has been consistent throughout all research in animal models of CKD [88].

Another key focus of research has been identifying specific downstream targets of the canonical Wnt signalling pathway in relation to kidney disease. Evidence has implicated multiple downstream targets of β-catenin in kidney injury and fibrotic CKD, including matrix metalloproteinase-7 (MMP-7) and plasminogen activator inhibitor-1 (PAI-1) [91]. MMP-7, a target gene of Wnt signalling with a role in epithelial repair and host defence, is usually expressed at low levels [91,93]. Upon kidney injury, it has been demonstrated that MMP-7 is induced and its expression is in fact increased in diseases, such as PKD and obstructive nephropathy [93]. Levels of renal MMP-7 have been found to correlate with Wnt/β-catenin activity in several CKD models [97].

As well as this, PAI-1 which is a secreted acute-phase glycoprotein, is also usually produced in low levels. PAI-1 is a target gene of β-catenin signaling, since the promoter region of PAI-1 contains a TCF/LEF-binding site [98]. Upregulating β-catenin has been shown to stimulate PAI-1 expression, whilst inhibition of β-catenin signalling/disruption of the TCF/LEF-binding site counteracts this [98]. In chronic kidney diseases such as DN, membranous nephropathy, FSGS and crescentic glomerulonephritis, PAI-1 expression becomes upregulated [99]. This evidence highlights PAI-1 as an important downstream target of β-catenin signalling in renal injury.

Whilst β-catenin signalling has been implicated in podocyte dysfunction, recent evidence has suggested that it could also cause podocyte injury by inducing transient receptor potential cation channel 6 (TRPC6), a calcium channel localised to podocytes, through exposure to high glucose [100]. Further evidence found that high-glucose-induced podocyte lesions improved upon inhibition of the canonical Wnt signalling pathway via DKK1 treatment, which also inhibited TRPC6 expression [100]. It is key to note that TRPC6 missense mutations have been associated with FSGS and thus, this research could demonstrate another way by which the canonical Wnt signalling pathway could cause podocyte dysfunction [101]. It is also important to note that whilst several factors result in the activation of the canonical Wnt signalling pathway, there are a number of antagonists of Wnt signalling, including soluble Frizzled-related proteins (sFRPs), Wnt inhibitory factor and the family of DKK proteins. As mentioned previously, DKK proteins antagonise Wnt signalling by binding and internalising LRP5/6, however, antagonists like sFRPs and Wnt inhibitory factor work by binding to Wnt proteins and consequently preventing the binding of Wnt proteins to Fz receptors [57,91]. By using this information, researchers have found that inhibiting β-catenin activation using DKK1 ameliorates kidney fibrosis in mouse models [92]. Further supporting this argument is a study that revealed loss of Klotho, another antagonist of β-catenin which is usually expressed in adult kidneys, in mouse models of CKD [102]. It was found that when Klotho was expressed *in vivo*, fibrotic lesions and kidney injury bettered [102].

Surendran et al. found that after Unilateral Ureteral Obstruction (UUO), the expression of both sFRP4 and DKK1 increased, whilst the administration of recombinant sFRP4 attenuated renal fibrosis [93]. Whilst further investigations found that these data correlate with a faster decline in renal function, other research has displayed that a decrease in sFRP4 relates to an increase in β-catenin signalling and an increase in the progression of renal fibrosis [93,103]. Corresponding with this, it was found that an increase in sFRP4 led to a decrease in β-catenin signalling and a decrease in the progression of renal fibrosis [93,103]. Elevated levels of sFRP4 expression have also been observed in human patients of ADPKD and animal models of cystic kidney disease, including in NPHP, highlighting a relationship between the induction of sFRP4 expression and renal cystogenesis [104]. Depletion of another Wnt antagonist, Dapper3, which colocalises with Dishevelled (Dvl) and forms a complex with cytoplasmic proteins Axin, glycogen synthase kinase-3 (GSK3), casein kinase 1 (CK1) and β-catenin causes the accumulation of Dvl2 and β-catenin and aggravates the fibrotic phenotype after UUO [105,106]. Tan et al. have suggested that RAS genes also have TCF/LEF-binding sites and are thus regulated by β-catenin signalling [91]. The group’s data shows that RAS genes are upregulated by β-catenin in vitro and in vivo and that the upregulation can be blocked by β-catenin inhibitors, such as ICG-001 [91,107]. Other evidence has found that canonical Wnt signalling inhibition via XAV939, a tankyrase inhibitor that promotes β-catenin degradation and LGK974, a Porcupine inhibitor that blocks lipid modification of Wnt proteins, delays the progression of renal cystogenesis and reduces elevated levels of β-catenin and of Wnt target genes, Axin-2, c-Myc and cyclin D1 in mutant mice [108].

## 6. Links between Wnt Signalling, PKD and NPHP

Aside from the role of Wnt proteins in kidney development and function, various research findings exist that link Wnt signalling to PKD and NPHP, two of the greatest groups of inherited cystic kidney diseases (Figure 1). The dysregulation of β-catenin was associated with cystic kidney disease in mice and other ADPKD models [51,88,109,110]. In a study where *Pkd2* was knocked out from a renal collecting duct cell line, there was evidence of ciliogenesis defects, together with increased β-catenin and AXIN-2 and c-Myc expression [55,111,112]. Further supporting this, is evidence from a *Pkd2* KO mouse line where the mutant mouse presented with kidney cysts, whilst the kidney tissue also demonstrated elevated levels of active, nuclear and total β-catenin *(Ctnnb1)* proteins alongside Axin-2, c-Myc and cyclin D1 proteins [108]. In the same study, loss of one *Ctnnb1* allele rescued the renal cystic phenotype and reduced the increased levels of Wnt target genes Axin-2, c-Myc and cyclin D1 [108]. In another study, the conditional inactivation of APC, a tumour suppressor that forms a key part of the multiprotein β-catenin destruction complex in the canonical Wnt signalling pathway resulted in cystic renal neoplasia [109].

Further highlighting the link between PKD and canonical Wnt signalling is a study that investigates the association between Aquaporin-1 (AQP1), a water channel protein with expression in the kidney, and Wnt signalling [113]. Upon overexpressing AQP1 in a mammalian cell line, Madin-Darby Canine Kidney (MDCK), reduced expression of β-catenin and cyclin D1 was observed [113]. As well as this, the study highlighted an interaction between AQP1 and various canonical Wnt proteins including β-catenin, GSK3β, LRP6 and Axin1 [113]. A MDCK cyst model also demonstrated that the overexpression of AQP1 inhibited cystogenesis and increased MDCK cell branching [113]. Upon KO of AQP1 in a PKD mouse model, investigators observed an increase in kidney size and cyst formation, together with increased Wnt signalling activity in the kidney [113]. Thus, it can be concluded that there is a relationship between canonical Wnt signalling and PKD.

It is important to note that evidence has implicated the Polycystin complex as a Wnt receptor through research that found the binding of Wnt ligands to the extracellular domain of PKD1 [114]. This includes Wnt ligands that act predominantly through the canonical Wnt signalling pathway (Wnt3a), non-canonical Wnt signalling pathway (Wnt5a) or both (Wnt9b and Wnt4) [114]. As well as this, the study found that the mediation of pronephric tubule formation in *Xenopus* is regulated by Wnt9a, Pkd1, and Dvl2 [114]. This study demonstrated that in *Pkd1 ^-^/^-^* cells, a downregulation of Dvl2 was observed [114]. Further experimental studies revealed that Dvl2 co-immunoprecipitated with Pkd1, implying that direct binding occurs. Additionally, knockdown of Dvl2 and Pkd1 in *Xenopus* resulted in similar cystic kidney phenotypes [114]. Furthermore, a study performed a few years prior revealed that *Xenopus* embryos injected with mutant Dvl2 presented broader wolffian ducts (~1.5x wider) and that the concertina-shaped ducts curved towards the cloaca instead of running in a straight line [115].

Inversin, a key protein encoded by *NPHP2*, is proposed to act as a molecular switch between the canonical and non-canonical Wnt/PCP signalling pathway [116]. It has been suggested that Inversin forms a complex with Dvl and prevents the action of the canonical Wnt signalling pathway upstream of the β-catenin degradation complex [116]. Phenotypes described in *Fz8^-/-^ Xenopus* resemble phenotypes exhibited when Inversin was depleted in other animal models [117]. It was also postulated that via the non-canonical pathway, Inversin colocalised with Dvl in the membranes of polarised renal epithelial cells. It was demonstrated that, in response to Fz receptors, Inversin induced the translocation of Dvl, and that during pronephros development, Inversin acted downstream of Fz8 induction. Furthermore, it was revealed that the overexpression of Inversin in *Fz8* depleted *Xenopus* was able to rescue defects during pronephros development [117]. Kidney defects and cyst formation in the pancreas and kidneys were phenotypes described in *inv^-/-^* mice [118]. Evidence showed that Inversin localised to the primary cilium in mouse embryonic fibroblast (MEFs) derived from *inv^-/-^* mice. The colocalisation of Fz3 was detected in the primary cilium of *inv^-/-^ and inv^+/+^* mice MEFs, implying that the induction of receptors in the cilium may regulate Wnt signalling and that Fz3 could be independent of inversin ciliary targeting [118].

The collective phenotypes exhibited when NPHP2 is disrupted include basement membrane abnormalities, renal interstitial fibrosis, kidney enlargement and cyst development [119]. The deletion of exons 3-11 of *Nphp2* in mice resulted in enlarged kidneys with widespread cyst formation alongside *situs inversus* and dysplasia of pancreatic islet cells [119]. The comparison of histological studies performed on *inv/inv* mouse and infantile NPHP2 samples revealed characteristic phenotypes commonly present in NPHP, e.g., interstitial fibrosis, tubular cell atrophy, and tubular cysts. In addition, the identified features resemble those observed in ADPKD; such as enlarged kidney and abnormalities in the basement membrane. Interestingly, this study demonstrated that *NPHP2* is a causative gene which causes NPHP-2 with or without *situs inversus*. Likewise, cystic phenotypes were identified in *inv* mutant zebrafish. Immunoblotting confirmed the co-precipitation of Inv and nephrocystin, suggesting direct binding. Similarly, a study conducted by Watanabe revealed that transgenic mice expressing a fusion of *inv* and Green fluorescence protein were able to rescue laterality defects and PKD in homozygous mutant *Inv* mice [120]. Moreover, evidence has shown a decrease in Inv in IgAN patients, alongside an increase in β-catenin, indicating the hyperactivation of the canonical Wnt signalling pathway [53].

A clinical study performed on 43 families with infantile NPHP revealed that *NPHP2* was mutated in 31% of cases and *NPHP3* in 16% of cases [121]. The following phenotypes were associated with *NPHP2* mutations: diverse kidney size, cerebellar atrophy, retinitis pigmentosa, and mild liver fibrosis [122]. Furthermore, Nphp4 can negatively regulate canonical Wnt signalling by interacting with Inv and Dvl to regulate the stability of Dvl [123]. Histological analysis revealed that tubular atrophy was present alongside thickened tubular basement membranes, diffuse interstitial fibrosis and cyst formation at the cortico–medullary junction in a *Nphp4* zebrafish model and cultured mammalian kidney cells [124].

It thus becomes apparent from the above that most of the evidence linking PKD to Wnt signalling is focused on canonical Wnt signalling (Figure 2). Such involvement is possibly mediated through upstream Wnt components and has an effect on β-catenin transcription. This does not mean that there is no involvement of non-canonical Wnt signalling in PKD, rather that it is hard to study non-canonical Wnt signalling in the context of inherited kidney disease, due to the lack of good PCP antibodies and limited samples or fully representative animal models of these diseases. Nevertheless, recent studies have accumulated evidence on dysregulated non-canonical Wnt signalling in PKD and it remains unclear as to whether this is a result of a shift in balance between canonical and non-canonical Wnt signalling or two separate events on both cascades.

On the other hand, NPHP has mostly been associated with changes in non-canonical Wnt signalling, with an initial hypothesis suggesting that the primary cilium acts as a switch that favours non-canonical Wnt signalling. Recent findings have contested this hypothesis, as cilia have now been shown to facilitate both canonical and non-canonical Wnt signalling. Nevertheless, increasing evidence is being acquired on the involvement of Wnt signalling in CKD. Work on animal models as well as human samples has demonstrated mis-regulated Wnt signalling in many genetically inherited chronic kidney diseases. This information could shed novel insights into kidney disease mechanisms and may hold potential therapeutic implications for this big group of diseases that overall pose a great burden on patients’ survival and well-being worldwide.

## Figures and Tables

**Figure 1 genes-11-00496-f001:**
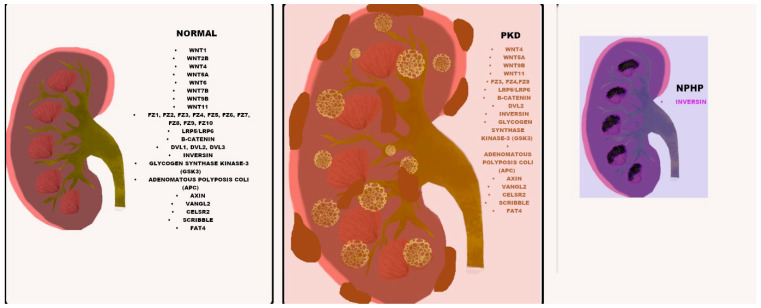
Wnt component expression in normal, PKD and NPHP kidneys. The listed Wnt components (black) are expressed in all three normal, PKD and NPHP kidneys, with differential expression in PKD indicated in orange and differential expression in NPHP shown in purple.

**Figure 2 genes-11-00496-f002:**
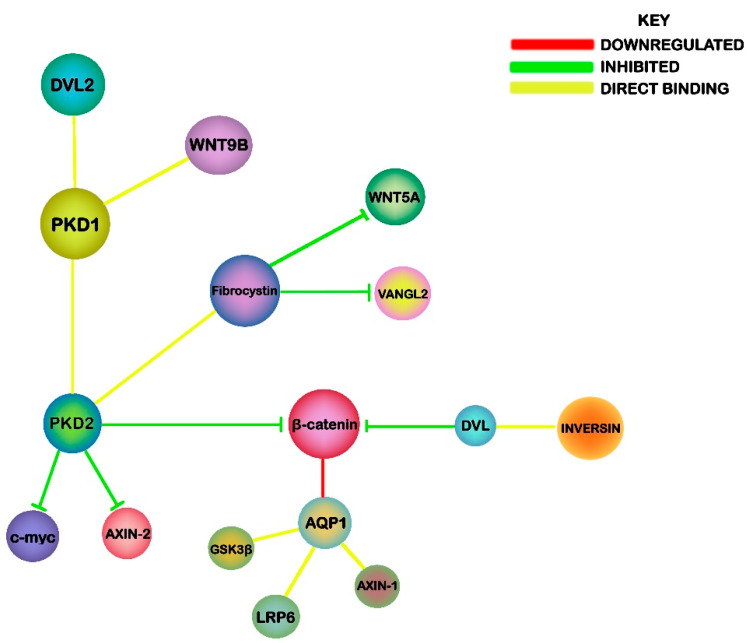
An illustration of the potential mechanisms linking Wnt signaling, PKD and NPHP. The associations highlight the relationship of PKD1 and PKD2 with canonical Wnt components, while Fibrocystin and Inversin are linked to PCP signaling.

**Table 1 genes-11-00496-t001:** A brief summary, including symptoms and associated genes of the reported genetically inherited chronic kidney diseases.

Disease	Brief Description	Associated Genes	References
**Autosomal Dominant Polycystic Kidney Disease (ADPKD)**	- Autosomal dominant - Affects around 12.5 million people in the world. - Symptoms include renal and hepatic cysts, hypertension, urinary tract infections.	- *PKD1* - *PKD2*	[2,3,4,5]
**Autosomal Recessive Polycystic Kidney Disease (ARPKD)**	- Autosomal recessive - Incidence rate: 1/20,000 - Affects neonates and infants. - Symptoms include enlarged, cystic kidneys, oligohydramnios, hepatic fibrosis.	- *PKHD1* - *DZIP1L1*	[6,7,8,9,10]
**Nephronophthisis (NPHP)**	- Autosomal recessive - Categorised into infantile, juvenile and adolescent NPHP - Symptoms include polyuria, polydipsia, anaemia, reduced kidney size and cysts in the corticomedullary area.	- *NPHP1 – NPHP20* - *AHI1*	[11,12,13,14,15,16]
**Joubert Syndrome (JS)**	- Autosomal recessive - 20%-30% of JS patients develop NPHP - Symptoms include hypotonia, hyperpnea, abnormal eye movements, delays in developmental abilities.	- *NPHP1* - *AHI1* - *TMEM67* - *CEP290* - *RPGRIP1L*	[16,17,18,19,20]
**Meckel-Gruber syndrome (MGS)**	- 100% mortality rate - Symptoms include polycystic kidneys, polydactyly and occipital encephalocele. - Defective cilia/Wnt signaling	- *NPHP3* - *CEP290* - *RPGRIP1L*	[16,17,18,19,21,22,23]
**IgA nephropathy (IgAN)**	- Incidence rate of 2.5/100,000 - Clinical manifestations include synpharyngitic macroscopic haematuria, proteinuria. - Varying prevalence based on ethnicity and gender observed with a possible increased risk in familial relations, indicating a genetic basis for IgAN - Diagnosis typically dependent on presence of granular deposition of IgA in mesangium by immunofluorescent analysis on kidney biopsy samples	- Risk alleles in the HLA region at chromosomes 6p21 and 1q32.	[24,25,26,27,28]
**Focal and segmental glomerulosclerosis (FSGS)**	- Incidence rate of 0.8/100,000 per annum, worldwide - Presentation of podocyte lesions - Can be categorised into idiopathic (primary) FSGS and secondary FSGS - Can also be categorised based on cause of disease and morphological appearance of lesions - Symptoms include oedema, proteinuria, microscopic haematuria, hypalbuminaemia and hypertension.	- *APOL1*	[29,30,31,32,33,34,35,36]

**Table 2 genes-11-00496-t002:** Wnt components have roles in various chronic kidney diseases (CKDs).

Wnt Component	Disease	Model	References
WNT1	FSGS	Human	[42]
WNT4	PKD, Membranous Nephropathy	Human, mouse	[43,44]
WNT5A	ARPKD, ESRD	Human	[45,46,47]
WNT9B	PKD	Mouse	[48,49]
WNT11	PKD	Mouse	[50]
LRP5	ADPKD	Human, cell line (HEK293)	[51]
LRP6	PKD	Mouse	[52]
β-catenin	ADPKD, IgAN, FSGS	Human, mouse, cell line (renal collecting duct cell line)	[42,53,54,55]
NPHP2/Inversin	ADPKD, IgAN, NPHP	Human	[53]
VANGL2	ARPKD	Human	[47]
CELSR2	JS	Human	[56]
SCRIBBLE	ARPKD	Human	[47]

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
