# Peer review of "The Role of Wnt Signalling in Chronic Kidney Disease (CKD)"

_genes, 2020, doi:10.3390/genes11050496_

Round 1

Reviewer 1 Report

The review manuscript by Malik et al. illustrates the current knowledge on the role of Wnt signalling in Chronic Kidney Disease. The paper provides a comprehensive overview of the current knowledge on the related topic and it is quite well-written and structured. Most of the cited literature in the paper is valuable and provide important insight into the mechanisms underlying the links between canonical and non-canonical Wnt signalling and this category of diseases. Overall, I think this manuscript could be suitable for publication in Genes; however, some comments should be addressed, as suggested below:

- A Table with the description of the reported diseases and known involved genes could help the reading of the literature data explained in the section 2 and should be added to the manuscript.

- Figure 1 should be provided with a higher resolution; moreover, it should be improved with a readable list of Wnt components by increasing the font size.

- The conclusion paragraph (section 6) should be expanded with a more critical discussion of the link between Wnt signaling components, PKD and NPHP. Additionally, this section would benefit of a further figure depicting possible involved mechanisms.

- A list of abbreviations is required.

- Reference 100: the year of publication is 2018 and not 2017.

- Some minor spelling errors and punctuation should be corrected throughout the text.

Author Response

Dear Editor,

We thank the reviewer for their comments, which have been addressed as tracked changes in the manuscript as described below.

- A Table with the description of the reported diseases and known involved genes could help the reading of the literature data explained in the section 2 and should be added to the manuscript.

This has now been provided as Table 1 in a separate file attachment.

- Figure 1 should be provided with a higher resolution; moreover, it should be improved with a readable list of Wnt components by increasing the font size.

A higher resolution of figure 1 has been provided and the fonts have been amended.

- The conclusion paragraph (section 6) should be expanded with a more critical discussion of the link between Wnt signaling components, PKD and NPHP. Additionally, this section would benefit of a further figure depicting possible involved mechanisms.

Additional discussion of the links between Wnt, PKD and NPHP has been added (lines 432-449). The newly created figure 2 illustrates potential mechanisms linking Wnt signalling, PKD and NPHP.

- A list of abbreviations is required.

A list of abbreviations has been included as a separate file.

- Reference 100: the year of publication is 2018 and not 2017.

This has now been changed to 2018.

- Some minor spelling errors and punctuation should be corrected throughout the text.

The text has been proofread and any spelling and punctuation errors have been corrected and they can be seen as tracked changes.

We hope that the reviewers’ comments have been successfully addressed and we look forward to receiving your response.

Kind regards,

Evi Goggolidou

Reviewer 2 Report

This manuscript is a great review to illustrate the role of both canonical Wnt signaling and non-canonical Wnt signaling in chronic kidney diseases. The authors detailed the basic information and clinic relevance about the interaction between Wnt signaling and CDK with a considerable amount of references, which will present the readers with a straightforward and thorough report in this scientific field.

Author Response

We thank the reviewer for their comments, it is good to hear that the manuscript represents a thorough review of the field.